# The Efficacy of an Immersive Virtual Reality Exergame Incorporating an Adaptive Cable Resistance System on Fitness and Cardiometabolic Measures: A 12-Week Randomized Controlled Trial

**DOI:** 10.3390/ijerph20010210

**Published:** 2022-12-23

**Authors:** Mitchell S. Mologne, Jonathan Hu, Erik Carrillo, David Gomez, Trent Yamamoto, Stevin Lu, Jonathan D. Browne, Brett A. Dolezal

**Affiliations:** 1Airway & UCFit Digital Health-Exercise Physiology Research Laboratory, David Geffen School of Medicine, University of California Los Angeles, Los Angeles, CA 90024, USA; 2John A Burns School of Medicine, University of Hawaii, Honolulu, HI 96813, USA; 3Keck School of Medicine, University of Southern California, Los Angeles, CA 90033, USA; 4Creighton School of Medicine, Omaha, NE 68178, USA; 5School of Medicine, California University of Science and Medicine, Colton, CA 92324, USA

**Keywords:** exergaming, immersive virtual reality, exercise, strength training

## Abstract

Exergaming, combining elements of video game into the realm of exercise, has recently incorporated immersive virtual reality (IVR) with resistance training. Thirty-two participants (14 females, mean age = 24.3) were randomized to IVR or self-directed control group (SELF) and worked out thrice weekly for 12 weeks (for 36 sessions). The IVR group spent 14 fewer minutes per session (*p* < 0.001) while reporting the sessions “enjoyable’. Compared to SELF, the IVR group had significantly greater improvement in changes from baseline to post-training in upper-and-lower muscular strength (1-RM) and muscular endurance (85% 1-RM) (14.3 kg vs. 10.0 kg for 1-RM upper, 28.6 kg vs. 22.5 kg for 1-RM lower, 2.6 reps vs. 1.9 reps for 85% 1-RM of upper, 2.7 vs. 2.0 reps for 85% 1-RM of lower, all *p* < 0.001), peak leg power (1424 vs. 865 W, *p* < 0.001), body fat% (−3.7% vs. −1.9%, *p* < 0.001), heart rate variability (4.3 vs. 1.8 ms, *p* < 0.001), rVO2max (3.28 vs. 0.89 mL/min/kg, *p* < 0.001) with decreased systolic BP (−0.4 vs. −2.3 mmHg, *p* < 0.001), and level of perceived exertion during workouts (RPE 14 vs. 16, *p* < 0.001). With its high-paced and action-filled gaming coupled with superior fitness and cardiometabolic outcomes, this IVR exergaming platform should be considered as another exercise modality for performance and health-related training.

## 1. Introduction

“Exergaming”, the portmanteau term coined for the coupling of exercise training and video game, has gained considerable popularity in the last few decades, notably with systems such as Dance Revolution, Wii Fit, and Xbox 360 Kinect [1,2]. By combining the competitive aspects and enjoyability of video games with physical activity, exergaming systems may attenuate perceived exertion and increase motivation while maintaining a high level of energy expenditure [2,3]. More recently, exergaming has begun to incorporate immersive virtual reality (IVR) elements to further enhance user experience [4]. This nascent technology produces a 3D space through both wall-mounted and wearable body sensors with head-mounted displays entailing stereoscopic vision and a large field of view, allowing participants to interact and to be ‘immersed’ in virtual environments in real-time, creating a feedback.

The use of IVR in exercise has shown an ability to elicit greater physiological and metabolic demand while maintaining or even minimizing perceived exertion by the user [1,5,6,7]. However, most prior studies on IVR exergaming have focused on aerobic exercise, neglecting other forms of exercise, such as the use of resistance training modalities. In a previously published study from this laboratory [8], using an IVR cable resistance exergaming system for an acute 30-min workout proved to elicit vigorous intensity exercise accompanied by high enjoyment levels and low perceived exertion from the participants. Moreover, the same platform that makes use of a unique servo-based electromagnetic dynamic resistance mechanism showed myoelectric activity similar to, and in some cases, better than conventional, compound strength exercises [9]. As it stands, to the best of our knowledge, there is no body of work analyzing extended adherence and the long-term chronic adaptations when comparing IVR resistance training to a traditional, self-directed cable resistance training protocol.

Exergaming may be the key to encouraging exercise participation for the general population. While lack of time has been cited as a common barrier to exercise [10,11,12,13], lack of motivation and enjoyment may also contribute to physical inactivity. Given the prevalence of video game use in younger generations [14], gamified exercise may be a reasonable alternative to traditional exercise training in gyms and at home.

This study aimed to assess changes in physical performance, body composition, cardiometabolic health, participant training efficiency, enjoyment, and perceived exertion during a 3-month workout protocol and to compare differences between groups assigned to either an IVR machine-directed exergaming system or a self-directed conventional resistance training control (SELF). We hypothesize that there will be greater improvements in body composition, fitness parameters, and cardiometabolic health after 3 months of IVR versus SELF-training. We also hypothesize participants will experience better training efficiency as well as greater enjoyment and lower perceived effort while maintaining higher levels of work after 3 months of IVR vs. SELF-training.

## 2. Materials and Methods

### 2.1. Participants

Recruitment of thirty-two participants at UCLA and the surrounding community met the following inclusion criteria: (i) apparently healthy men and women, (ii) 18–35 years of age, and (iii) history of exercising < 4 workouts/monthly the past 6 months. Exclusion criteria included: (i) significant medical diagnoses, including cardiovascular, pulmonary, musculoskeletal, or metabolic disorders that may limit the ability to exercise or increase the cardiovascular risk of exercising, and (ii) use of any drug or supplement known to enhance anabolic responses. All volunteers completed a pre-participation physical activity readiness questionnaire (PAR-Q) [15] and an exercise history questionnaire. The UCLA Institutional Review Board reviewed and approved the study, and all participants gave their written informed consent. This study was conducted according to international standards for sport and exercise science research.

### 2.2. Study Design

This was a 12-week, prospective, single-blinded, randomized control trial. Using a parallel research design, healthy, college-aged men and women volunteers (n = 32) with minimal resistance training experience were randomly allocated 1:1 (16 per group) into one of two groups: the intervention, IVR machine-directed (“IVR”) or the control, self-directed conventional resistance training (“SELF”), by an investigator independent of the recruitment of participants using an online-generated random number program. Allocation was concealed with the use of consecutively numbered envelopes (Figure 1). The participants worked out thrice weekly (i.e., a total of 36 sessions) for between 35–60 min with the primary objective of increasing muscle strength and lean body mass with a concomitant decrease in fat mass. A 12-week trial was selected to ensure a training adaptation from both research arms. Participants were asked to refrain from additional vigorous activity for the course of the study.

### 2.3. Interventions

All assessments and training were administered by trained research personnel under the direction of the lab director in the UC Fit Digital Health–Exercise Physiology Research Laboratory on UCLA’s campus. Both groups trained at their discretion and were asked to train only 3 times weekly, preferably with a rest day between sessions. Moreover, study participants were not prompted or enticed to work out by study personnel. Dietary intake and macronutrient portions were not controlled apart from the requirement of not starting a dietary supplement or weight loss/gain diet that might affect total and fat-free body mass.

### 2.4. Study Groups

#### 2.4.1. Intervention Group: BlackBox Immersive Virtual Reality (IVR)

The IVR-based cable resistance exergaming machine, Black Box VR (Black Box VR^®^, Boise, ID, USA), employed a servo-based electromagnetic adaptive resistance mechanism (Figure 2). The system also included a head-mounted-display (HMD) (Vive, Taipei, Taiwan), an automated support pad, and pair of resistance handles that adjust up and down on articulating carriages to automatically configure for all cable resistance exercises. The system’s HMD and wrist-worn sensors processed movement data to ensure proper syncing between the user’s actions and the IVR gameplay. 

The exergame was similar to a traditional, two-lane tower defense where the user’s goal was to protect their crystal by fending off enemy units while concurrently dealing damage to the opponent’s crystal. Six cable resistance exercises (i.e., lat pulldown, standing chest press, standing row, overhead press, stiff-leg deadlift, and squat) were linked to in-game attacks where each exercise corresponded to a unique attack move (Figure 3). The damage delivered was dependent on the cable resistance level. Per the manufacturer, the exergame system’s proprietary software adjusted the resistance when 12–14 repetitions were completed per set at 60–70% of the users automatically calculated the predicted one-rep maximum. In addition, the attack moves were based on elemental categories such as fire, water, and air, and could be selected to counter enemy actions (e.g., a squat performed a water attack, which is best used against fire enemies). Thus, the selection of individual exercises was determined by gameplay strategy, but the intensity and number of repetitions performed depended entirely on the user’s ability [8,9].

It should be noted that during the IVR exergaming session, each study participant freely chooses their exercise, exercise sequence, and ultimately the training volume. Because of this, these variables (i.e., exercise choice and sequence, sets, repetitions, and time-to-complete) were computed by the machine’s integrated computer, which subsequently resulted in a training regimen that was matched to a control participant for each of the 36 training sessions to prevent training dose–response bias in the outcome measures.

#### 2.4.2. Control Group: Self-Directed (SELF)

Matched participants received the training regimen to guide the SELF workouts. Self-directed training made use of a conventional, cable/pulley-based crossover machine using dual-selectorized weight stacks (Figure 1). The cable-pulley arms are spaced the same distance as the IVR machine and offer the same vertical adjustment options. Aside from the different training resistance (i.e., torque), the two machines could mimic near-identical movement patterns for the six resistance exercises making this an ideal control. An iPhone app (Airway & UCFit Digital Health-Exercise Physiology Research Laboratory, Los Angeles, CA, USA) tracked all sessions, including volume (i.e., sets multiplied by reps) of exercises performed, and was subsequently used to guide a SEL2F group participant’s workout.

### 2.5. Experimental Procedures: Baseline and Post-Measures

#### 2.5.1. Cardiometabolic Measures

Anthropometry: Body mass was measured in duplicate on a calibrated medical scale (accuracy ± 0.1 kg), and height was determined using a precision stadiometer (Seca, Hanover, MD, USA; accuracy ± 0.01 m). For mass, participants removed unnecessary clothing and accessories. For height, participants were instructed to stand as straight as possible with unshod feet flat on the floor.

Blood Pressure: Seated blood pressure was measured in duplicate on the bare left arm after participants had rested quietly for a minimum of 10 min using an automated blood pressure system (Omron, Vernon Hills, IL, USA). Participants were asked not to talk, engage their cell phones, or cross their legs during this time. Cuff sizes were adjusted to the participant, and the same arm was used in pre and post-measurements.

Resting Metabolic Rate: A portable metabolic analysis system (PNOĒ^®^, Palo Alto, CA, USA) was used to determine resting metabolic rate. The previously validated PNOĒ system was placed next to the participant as they lay in a semi-inclined position in a temperature-controlled (22 °C) room with dimmed lighting, absent from noise distraction [16]. Participants donned a standard facemask and head support (Hans Rudolph Inc.^®^, Shawnee, KS, USA) and breathed through a flow sensor that gas analyzer for measures of oxygen consumption (VO_2_) and carbon dioxide production (VCO_2_). Gas analyzers were calibrated before each assessment per manufacturer instructions. To allow participants to acclimate to the facemask and establish a resting baseline, RMR was measured for 5 min before data collection. Data was transmitted in real-time via Bluetooth to the system’s cloud storage platform for analysis.

Heart Rate Variability: The participants were fitted with a wrist-worn device and associated smartphone application (Biostrap USA LLC, Los Angeles, CA, USA) to capture their root-mean-square-of-successive-differences-between-normal-heartbeats (RMSSD)—a validated metric that determines the vagally mediated HRV response [17]. Using a proprietary PPG processing software (Wavelet wristband, Wavelet Health, Mountain View, CA, USA), the device captured a 60-s reading, producing HRV values with high signal quality. Participants were asked to avoid food, nutritional supplements (e.g., creatine), caffeine, alcohol, smoking, and heavy physical activity for 12 h before testing. Testing was performed in a comfortable, temperature-controlled (22 °C) room with dimmed lighting, absent from noise distraction. The same conditions were imposed for post-test measurements.

#### 2.5.2. Fitness Measures

Body Composition: Body fat percentage, fat mass, and fat-free mass was measured using a validated octipolar, multi-frequency, multi-segmental bioelectrical impedance analyzer (InBody Co., Seoul, Korea Republic) [18]. To ensure accuracy, participants adhered to standard pre-measurement BIA guidelines recommended by the American Society of Exercise Physiologists [19]. Briefly, the test was performed after at least three hours of fasting and voiding, with participants instructed to remain hydrated and not exercise 2 h before testing. After investigators explained the procedure, the participant stood upright with their feet on two metallic footpads while holding a handgrip with both hands. The instrument measured resistance and reactance using proprietary algorithms.

Muscle Strength and Endurance: Muscle strength was assessed by the 1-repetition maximum (1-RM) method for squat and bench press exercises. The 1-RM is the highest weight lifted through a full range of motion at the correct speed only once. Participants first warmed up on either a treadmill or cycle ergometer and did light stretching. Examiners allowed participants to practice the movement with no load before gradually adding weight and having them perform the first set with 6 to 8 repetitions. After one minute of rest, the load is increased, and the participant performed 3 to 4 repetitions. After one min rest, the participant performed 1 to 2 repetitions at a load estimated to be near a maximal effort. A final two-minute rest is given, the participant then attempts their 1-RM. For each 1-RM trial, participants attempted two repetitions. Muscle endurance was then measured as the number of repetitions to failure using 85% of baseline squat and chest press 1-RM values [20].

During the first and last training sessions (i.e., sessions 1 and 36), the aforementioned sequence to determine 3-RM’s was applied to each of the 6 resistance training exercises using the cable crossover machine.

Lower Body Peak Power: Leg peak power was estimated using a previously validated electronic jump mat (Probotics, Inc., Huntsville, AL, USA) [21]. Participants stood on a mat with feet at hip-width and then performed a countermovement jump for maximal height. Jump height was recorded with a handheld computer interfaced with the jump mat. Three trials were given with 30 s rest between trials. The best trial was used to calculate peak leg power using published equations that required jump height and the participant’s body mass. Jump height was determined from “hang-time”, defined as the time (s) from the feet leaving the mat to their return and the following equation: Ht = t^2^ * 1.227 where t is hang-time in seconds, and 1.227 is a constant derived from the acceleration of gravity [22].

Aerobic Performance: Aerobic capacity, VO_2_max, was determined via gas exchange using an incremental symptom-limited maximal treadmill exercise test. Standard procedures [23], including using individually determined work rate protocols that predict test completion within 8–12 min, were administered. Gas exchange was measured with a metabolic measurement system (the same portable PNOĒ system described in the RMR section) incorporating a flow sensor and discrete oxygen and carbon dioxide analyzers. Proprietary algorithms time-aligned flow and gas concentrations breath-by-breath and displayed 8-breath rolling averages for pulmonary minute ventilation (V_E_), oxygen uptake (VO_2_), and carbon dioxide output (VCO_2_). These data were continuously monitored and recorded during 3 min of baseline and throughout exercise and recovery. Heart rate (HR) was assessed via an affixed strap around the chest (Polar Electro Inc.^®^, Kempele, Finland). HR during the session was recorded in 15-s epochs and simultaneously time-aligned with the portable indirect calorimeter. Testing was conducted by trained and well-experienced personnel under established guidelines for cardiopulmonary exercise testing. Participants were tested wearing a T-shirt, shorts, and athletic footwear and encouraged to exercise to exhaustion. Maximum oxygen uptake was be taken as the highest VO_2_ achieved during a 15-s measurement interval.

Modified Sit-in-Reach Test: The modified sit-and-reach test is a standard measurement tool for evaluating hamstring and lower back flexibility. The test uses a 12-inch sit-and-reach box and the finger-to-box distance as the relative zero point. The participant sat on the floor with their buttocks, shoulders, and head in contact with the wall. The participant then extended the knees and placed the soles of his or her feet against the box. A yardstick was placed on top of the box with the zero end toward the participant. Keeping the head and shoulders in contact with the wall, the participant reached forward with one hand on top of the other, and the yardstick was positioned so that it touched the fingertips. This procedure established the relative zero point for each participant. The participant then reached forward slowly, sliding the fingers along the top of the yardstick. The score (distance in cm) was the most distant point on the yardstick contacted by the fingertips. An average of 3 trials were recorded, with the largest score being used for analysis. Participants will be allowed to rest for 30 sec between trials [24].

### 2.6. Questionnaires

Participants completed the following questionnaires via an iPhone app (Airway & UCFit Digital Health-Exercise Physiology Research Laboratory, Los Angeles, CA, USA):

Rating of Perceived Exertion (RPE) Scale: The Borg 6–20 scale was used to measure the rating of perceived exertion. The Borg scale allowed participants to rate their own perceived level of exertion on a scale from 6 (no exertion at all) to 20 (maximal exertion) [25]. RPE scores have been widely used for a half-century, and studies suggest these scores have a high correlation with heart rate.

Physical Activity Enjoyment Scale (PACES): Enjoyment of the workouts was measured using the Physical Activity Enjoyment Scale [26]. This questionnaire consisted of 16 items scored on a scale from 1 (strongly disagree) to 5 (strongly agree) to determine the participant’s level of physical activity enjoyment. A high overall mean score correlates with a high level of enjoyment. PACES results have shown acceptable internal consistency.

Simulator Sickness Questionnaire (SSQ): For the IVR group only, perceived motion sickness and cybersickness symptoms were assessed on a four-point scale (0–3) and categorized based on oculomotor discomfort, disorientation, and nausea [27].

### 2.7. Adherence and Compliance

Exercise adherence was recorded by taking attendance via a smartphone app during the 12-week protocol. Each participant was provided with a recommendation to exercise 3 days per week while attendance was evaluated as the actual number of sessions attended, divided by the recommended number of sessions (n = 36).

### 2.8. Statistical Analysis

Based on pilot testing and allowing for 15% missing data, a sample size of 32 participants (i.e., 16 per research arm) was determined to be sufficient to assess changes in fitness outcomes based on α = 0.05 and β = 0.20. All data were exported to IBM SPSS Statistics for Windows, version 24 (IBM Corp., Armonk, N.Y., USA) for analysis with an a priori α level of = 0.05, and all tests were two-tailed. Descriptive statistics are presented as mean (SD). Grubbs’ test was employed to detect potential outliers, and none were found. Before comparisons, all variables were assessed for normality via Shapiro–Wilk tests. Within-group (IVR vs. SELF) comparisons at baseline and after 12 weeks were made by paired t-tests and Wilcoxon signed-rank tests for normally and non-normally distributed variables, respectively. Changes between groups were analyzed by Welch’s independent t-tests (normal) or Mann–Whitney U tests (non-normal). Given that this is one of the first randomized trials testing an immersive virtual reality resistance training platform, we did not employ strict type 1 error control; however, we limited the number of main outcome measures and based our interpretation on the pattern of results seen for each domain rather than on individual statistical tests.

## 3. Results

Thirty-two study participants (14 females) had an average age of 24.3 ± 4.0, ranging from 19 to 33 years old, completed the 12-week training program without injuries or serious adverse events, although three participants in each group required an additional 1–2 weeks to complete the program due to minor illness or vacation. Training compliance for three sessions weekly for a total of 36 sessions was 100% for both groups. Although each session’s average training volume (i.e., exercises x sets x volume x weight) between groups did not differ during the study (SELF; 17,345 ± 2310 kg vs. IVR; 17,678 ± 3287 kg), the average training time per session was almost 14 min longer for the SELF vs. IVR group (44 ± 5.7 vs. 30 ± 0 min, *p* < 0.001).

### 3.1. Anthropometric, Fitness, and Cardiometabolic Results

#### 3.1.1. Anthropometric Measures

Anthropometric measures are described in Table 1. No differences existed between groups in age, height, and body mass at baseline and post-training. Body composition improved in both groups, though body fat % and absolute fat mass loss decreased almost two times more in the IVR group. 

#### 3.1.2. Fitness

Fitness measures are reported in Table 1. No differences existed between groups in any of the fitness measures at baseline. Both upper and lower body muscle strength and endurance improved significantly in all participants, with the IVR group having a significantly greater magnitude in all 1-RM and 3-RM (Figure 4). Peak lower body power also increased significantly in both groups; however, there was significantly greater improvement in the IVR group versus the SELF group.

Aerobic performance measures improved significantly in the IVR group only, although the SELF group was trending upward for the rVO_2_max.

Lower-back/hamstring modified sit-in-reach significantly improved in both groups throughout training, however, significantly at a much greater magnitude for the IVR group.

#### 3.1.3. Cardiometabolic

Cardiometabolic measures are reported in Table 1. At baseline, no differences existed between groups in blood pressure, RMR, and HRV. Only the IVR group showed a significant drop in systolic blood pressure, while both groups showed no change in diastolic blood pressure. Resting metabolic rate and heart rate variability significantly improved in both groups, although with an over 2-fold greater increase in the latter amongst the IVR group.

### 3.2. Questionnaires

IVR participants classified their training sessions to be ‘‘somewhat hard-to-hard’’ with an RPE score of 14 ± 1 while indicating the sessions to be ‘‘enjoyable’’ with a PACES average score of 4.45 ± 0.22. These participants did not report any cybersickness symptoms, demonstrating an average total SSQ score of 25.2 ± 23.0. SELF participants classified their training sessions to be “hard to very hard” with an RPE score of 16 ± 1.

## 4. Discussion

Given the shown benefits of exercise, it is important from a public health lens to get more of the general population to exercise [28]. In comparison to our prior two publications regarding the acute effects of immersive virtual reality (IVR) on exercise, this study analyzed the long-term physiological and cardiometabolic adaptations that IVR exergaming may elicit and its possible role in keeping people active. To our knowledge, this study is one of the first randomized, double-blinded controlled trials to compare a “gamified” IVR cable resistance system with a self-guided cable resistance training regimen. This 12-week exergaming experience proved to be more effective than traditional, volume-matched resistance training in improving every outcome measure, including body composition, cardiometabolic metrics, and muscle strength, power, and endurance.

Many prior studies have shown the efficacy of resistance training in improving body composition, namely through decreased body fat and maintained/gained lean muscle [29]. Interestingly, while both groups experienced a significant reduction in fat loss and body fat %, the IVR group’s losses were nearly two-fold greater than the SELF’s. We posit that the structure and gamification of the IVR system, which adds an element of high-intensity interval/aerobic training in conjunction with resistance training [8], may contribute to much of this result. In the game, players were incentivized to perform more cable exercises to perform “attack moves” that protect their tower. Instead of passive recovery (i.e., sitting) in between sets, players often needed to deploy in-game support “heroes”, which could only be done by completing a series of virtual agility and aerobic movements such as karate chops. This active recovery may elicit an aerobic component to the IVR workouts that were not experienced by the SELF group, hence leading to higher energy expenditures and greater fat loss [30,31]. Ultimately, the demands of the game encouraged little ‘downtime’ in between sets of cable exercises—a stark contrast to typical weight training in which stationary rest periods are often practiced. These results are comparable to the work by Villanueva et al., which observed greater decreases in body fat % when participants trained with shorter rest periods [32]. Participants in the IVR group also experienced significantly better improvements in the VO2 max and rVO2max (200% higher increase and 269% higher increase, respectively) that were not seen in the SELF group. This improvement in aerobic outcomes is also a likely product of the gamification described herein. Moreover, various review articles report that resistance training may lead to increased VO2max, likely through increased muscle mass and blood flow, and possibly, through decreased rest between sets [33,34].

Participants also experienced cardiometabolic adaptations, as both groups saw improvements in RMR and HRV-RMSSD, with the IVR group experiencing even heightened improvements. Similarly, the IVR group experienced a significant decrease in systolic blood pressure. The increased RMR is likely a result of an increase in lean body mass. Our study is similar to past research that has shown extended periods of resistance training elicit an increase in RMR that is typically accompanied by an increase in fat-free mass [35,36]. While both groups experienced an increase in RMR, the IVR group saw an almost 2.0x larger increase in their fat-free mass that accompanied larger increases in their RMR compared to the SELF group. It has been well-established that resistance training may provide decreases in systolic blood pressure. A meta-analysis by Cornelissen et al. found that resistance training for over 4 weeks elicits decreases in systolic blood pressure, decreasing an average of 3.9 mmHg [33]. In our study, only the IVR group experienced a significant decrease in systolic blood pressure by about 2.3 mmHg, which may be attributable to the greater weight loss experienced by IVR participants. This trend was also observed by Wing et al. who saw significant decreases in systolic blood pressure only once participants lost between 2% and 5% body fat [37]. The IVR group lost an average of 2.4% of body fat, which corresponded to only a significant decrease in systolic blood pressure. The SELF group lost an average of 1.9% and did not see any significant decreases in blood pressure. 

The IVR group showed greater benefit in overall muscular performance metrics, displaying a nearly 1.5x greater increase in 1 and 3-RM strength measures and muscle endurance, as depicted in Figure 3, and a nearly 2x greater increase peak lower body power. As described in previous studies [8,9], the IVR exergaming system’s training algorithm is designed to adjust the cable resistance so that the user is trained to “near muscle failure” each time. More specifically, the user’s force generation, range of motion, and speed of execution are all calibrated so that users selecting low, medium, and high-intensity workouts achieve near muscle failure at the 9–12, 13–16, and 22–25 repetition ranges, respectively. Since near and complete muscle failure exercises elicit comparable muscle activation [38], we posit that the IVR system’s optimization of near-muscle failure exercise effectively encourages intense workouts while minimizing the risks of excessive mechanical stress involved in training to complete failure. It was unknown why peak leg power increased more in the IVR group, but it likely stems from the heightened strength gains seen in the lower body. Furthermore, having an algorithm control the progressive overload over the 36 sessions may have provided greater training efficacy compared to standard cable resistance machines where users ‘pick’ their weight. This may be similar to prior studies that have shown a higher efficacy in workouts when having guidance (i.e., a personal trainer) as compared to self-directed training [39].

IVR exergaming provided participants with a more time-efficient workout than the SELF group. While all participants performed all 36 training sessions with non-significant differences in training volume, it took the experimental group approximately 14 fewer minutes, or 31.8% less workout time per session. Some research indicates that muscle strength is not influenced by time as long as the volume is equal [40]. Similarly, in a review of duration, frequency, and length, the greatest improvements were seen when exercise participants reached levels within 90–100% of their VO2max [41]. With this in mind, this study found that the equal volume for the IVR group, coupled with its heightened cardiovascular and anaerobic demand as a result of its gameplay, would lead to significantly greater results in a more efficient manner. Finally, prior studies have shown that high-intensity interval workouts, while time-efficient, are often met with low levels of satisfaction and enjoyment [42]. However, exergaming, especially those using immersive virtual reality elements like in this study, may combat this through the gamification of the workout. It is known that this additive entertainment component to users, essentially diverting their attention to the gameplay and away from the exercise effort, will intensify their ‘real’ effort [2,3]. The IVR group reported high level of enjoyability in their sessions with an average PACES score of 4.45 ± 0.22, while still finding the workouts to be less strenuous than the control group. In fact, SELF participants classified their sessions to be “hard to very hard” with an RPE score of 16 ± 1 while IVR only reported an RPE of 14 ± 1, indicating their sessions to be “somewhat hard-to-hard”. Hence, IVR exergaming users may experience the time-efficient benefits of high-intensity training without the negatives that accompany it. 

## 5. Limitations

While this study contributes to the existing literature on exergaming, its attenuation of perceived exertion, and its high levels of enjoyment, it is not without limitations. First, the study only aimed to explore differences of an IVR exergaming intervention and a traditional cable resistance protocol. As such, gamification and immersion were variables included only in the experimental IVR group, and differences could not be made based off of only one of the variables. A recent article by Xu and colleagues explored the use of virtual reality system versus a non-VR motion based system in exercise and found better performance in virtual reality [43]. A future study would be beneficial to similarly explore the effects of immersion while controlling for gamification, using the same exergaming system. Secondly, the study cohort consisted of young college-aged individuals, and the results of this study may not be generalized to other demographics. Similarly, both cohorts were filled with enthusiastic, highly motivated individuals who all completed the workout protocol timely and strongly adhered to it, which may not represent the general population. Another limitation is that participants on occasion needed to scheduled sessions ahead of time, which is not necessarily a reflection of “free-living” exercise. Lastly, as the IVR system allows players to choose their exercises as a means of strategy, there may have been variability caused by inconsistencies in the frequency of lifts between participants in the experimental group. Hence, more long-term studies will be needed to show the efficacy of IVR.

## 6. Conclusions

This randomized controlled study demonstrates that an immersive virtual reality exergaming system with the goal of improving anthropometric, cardiometabolic, and fitness parameters such as body fat percentage, lean body mass, 1-RM and 3-RM, and rVO2max, among others, may lend itself to be more superior and time-efficient when compared to a traditional, cable-resistance protocol. Our findings suggest that this IVR exergaming platform compared to traditional resistance training may be a better modality for individuals to achieve better fitness and cardiometabolic outcomes while reporting high enjoyability and attenuated perceived exertion.

## Figures and Tables

**Figure 1 ijerph-20-00210-f001:**
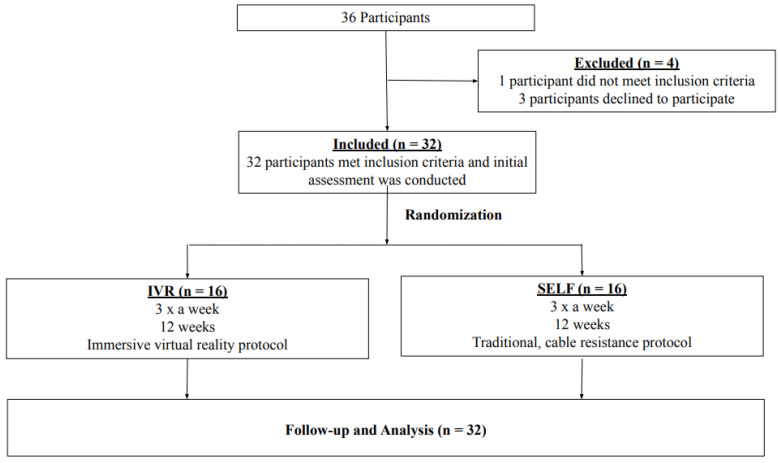
CONSORT diagram showing participant flow through the study. IVR, Intervention; SELF, Control.

**Figure 2 ijerph-20-00210-f002:**
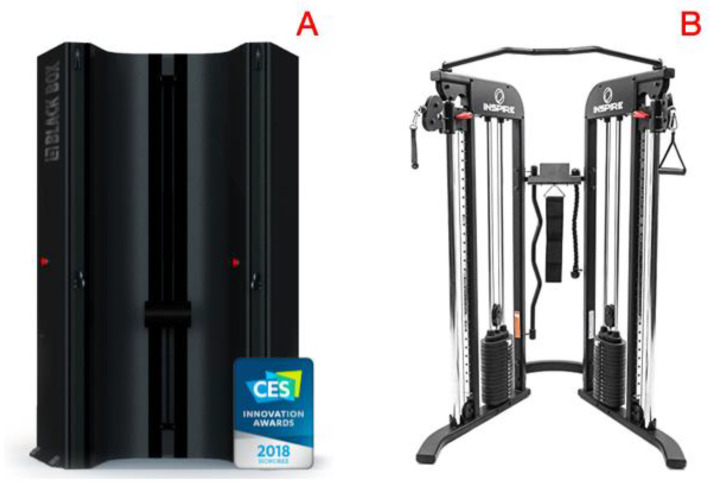
Blackbox VR^®^ servo-based system used by IVR (**A**) compared to conventional cable/pulley-based crossover machine using dual-selectorized weights (**B**) used by SELF.

**Figure 3 ijerph-20-00210-f003:**
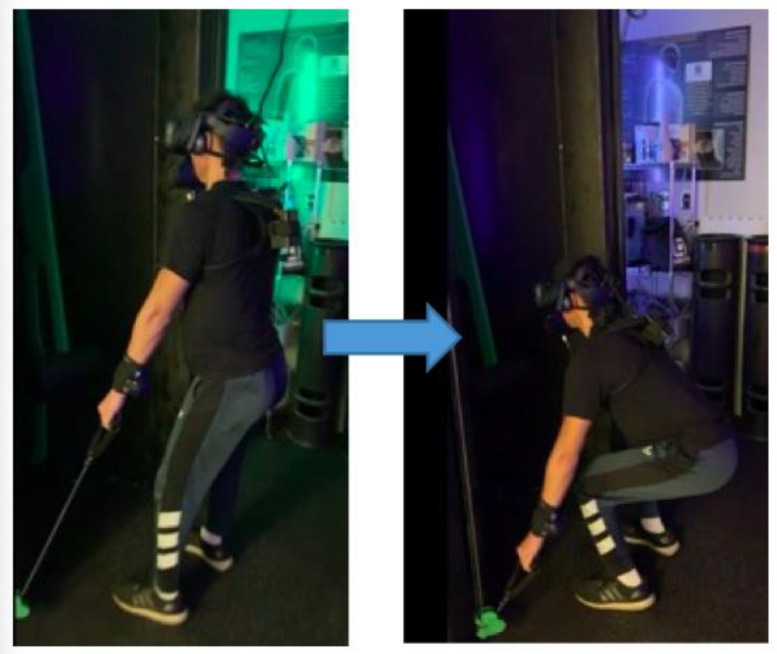
Participant donning an IVR head-mounted-display while performing a cable-based front squat using the Blackbox VR^®^ system.

**Figure 4 ijerph-20-00210-f004:**
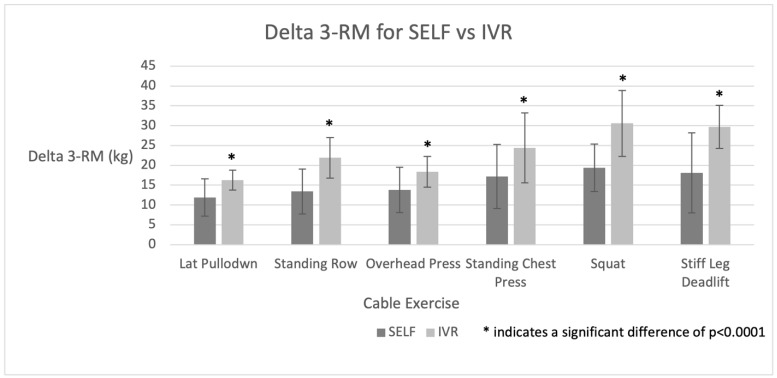
Change in Baseline to 12-week Post-intervention 3-RM for Various Cable Exercises for IVR and SELF Groups.

**Table 1 ijerph-20-00210-t001:** Anthropometric, fitness, and cardiometabolic measures at baseline and after 12 weeks for the IVR and SELF groups.

Measures	SELF (n = 16; 7 Females)				BBIVR (n = 16; 7 Females)				*p*-between
	Baseline	12 Weeks	Change	*p*-within	Baseline	12 Weeks	Change	*p*-within	
**Anthropometric**									
**Height (cm)**	172 (9.7)	-	-	-	168 (8.9)	-	-	-	0.873
**Body mass (kg)**	69.8 (8.2)	70.3 (8.1)	0.5 (1.1)	0.430	68.8 (12.7)	69.7 (12.3)	0.9 (1.2)	0.211	0.118
**Body fat (%)**	18.3 (2.9)	16.4 (2.8)	−1.9 (1.0)	**<0.001**	19.9 (3.4)	16.1 (3.0)	−3.8 (1.2)	**<0.001**	**<0.001**
**Fat mass (kg)**	12.8 (2.8)	11.6 (2.4)	−1.2 (0.8)	**0.004**	13.7 (3.7)	11.3 (3)	−2.4 (1.1)	**<0.001**	**<0.001**
**Fat-free mass (kg)**	57.0 (6.3)	58.8 (6.8)	1.8 (1.2)	**0.003**	55.1 (10.1)	58.5 (10.4)	3.4 (0.9)	**<0.001**	**<0.001**
**Fitness**									
**CP 1-RM (kg)**	50.5 (15.5)	60.5 (15.5)	10.0 (2.8)	**<0.001**	51.7 (17.8)	66.0 (18.7)	14.3 (2.0)	**<0.001**	**<0.001**
**SP 1-RM (kg)**	63.7 (17.7)	86.2 (19.3)	22.5 (5.7)	**<0.001**	64.3 (15.3)	92.9 (20.0)	28.6 (7.3)	**<0.001**	**<0.001**
**CP 85% 1-RM (reps)**	3.6 (0.8)	5.4 (1.1)	1.9 (0.7)	**<0.001**	3.5 (0.7)	6.1 (1.1)	2.6 (0.8)	**<0.001**	**<0.001**
**SP 85% 1-RM (reps)**	7.2 (2.6)	9.2 (4.1)	2.0 (3.7)	**<0.001**	4.1 (1.0)	6.8 (1.1)	2.7 (0.7)	**<0.001**	**<0.001**
**Leg power_peak_ (W)**	4945 (385)	5810 (447)	865 (194)	**<0.001**	4987 (469)	6411 (314)	1424 (326)	**<0.001**	**<0.001**
**VO_2_max (L/min)**	2.73 (0.19)	2.82 (0.22)	0.08 (0.09)	0.789	2.60 (0.36)	2.87 (0.35)	0.27 (0.10)	**<0.001**	**<0.001**
**rVO_2_max (mL/min/kg)**	39.38 (3.52)	40.26 (3.54)	0.89 (1.33)	0.552	38.38 (5.22)	41.66 (4.88)	3.28 (1.24)	**<0.001**	**<0.001**
*** Sit-in-reach (cm)**	33 (4.8)	36 (5.2)	3.0 (1.1)	**0.008**	32 (4.9)	37 (5.3)	5.0 (0.8)	**<0.001**	**<0.001**
**Cardiometabolic**									
**Systolic-BP (mmHg)**	121.6 (2.7)	121.2 (2.7)	−0.4 (1.4)	0.422	122.2 (3.8)	119.9 (2.7)	−2.3 (1.7)	**0.005**	**<0.001**
**Diastolic-BP (mmHg)**	80.7 (2.3)	80.3 (2.0)	−0.4 (1.6)	0.732	80.3 (3.3)	80.3 (2.2)	0.0 (1.6)	0.798	0.843
**RMR (kcal)**	2028 (129)	2130 (125)	102 (68)	**0.005**	2044 (138)	2176 (143)	132 (64)	**0.005**	**0.006**
**HRV-rMSSD (ms)**	37.9 (2.7)	39.7 (2.4)	1.8 (1.1)	**0.008**	38.7 (3.2)	43.0 (3.5)	4.3 (1.7)	**<0.001**	**<0.001**

Values are mean (SD). No significant differences were observed between groups at baseline. CP = chest press; SP = squat press; 1-RM = 1-repetition maximum; VO_2_max = maximum oxygen uptake; rVO_2_max = maximum oxygen uptake normalized by body mass; BP = blood pressure; RMR = resting metabolic rate; HRV-rMSSD = heart rate variability-root mean square of successive differences between normal heartbeats; * Modified sit-in-reach test was used.

## Data Availability

The data presented in this study are available on request from the corresponding author.

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
