# Peer review of "The Efficacy of an Immersive Virtual Reality Exergame Incorporating an Adaptive Cable Resistance System on Fitness and Cardiometabolic Measures: A 12-Week Randomized Controlled Trial"

_ijerph, 2022, doi:10.3390/ijerph20010210_

Round 1

Reviewer 1 Report

. I have reviewed the manuscript entitled “The Efficacy of an Immersive Virtual Reality Exergame Incorporating an Adaptive Cable Resistance System on Fitness and Cardiometabolic Measures: A 12-Week Randomized Controlled Trial” for consideration to be published in International Journal of Environmental Research and Public Health.

It is a very interesting study that probably will encourage another authors to perform further studies. Congratulations to the authors.
However, prior to its potential publication, there are some questions that should be reviewed in my opinion.

1. On what outcome did the authors base the randomization?

2. The authors should review the list of references, specifically references 10, 20, 33 and 34, since they contain errors.

Reviewer 2 Report

In this paper, the authors present an experiment that compared a normal workout with cable resistance against using an immersive VR exergame over a 12-week period. This is an interesting work and provides further evidence to support the use of immersive media to encourage more active and healthier living.

The paper is well-written and easy to follow and will be of interest to researchers working with VR exergames.

There is one aspect that the authors did not mention much, but it is important when reporting the conclusions. This aspect is about the fact that there are two factors in the immersive VR setting that are different from the traditional workout: (1) the technology (VR) and (2) gaming elements. I wonder if the authors considered adding gaming elements to the traditional workout. For example, it is possible to use a normal TV screen and have the same gaming environment given in the traditional platform to participants. Because of these two factors put together, it is not clear whether it is because of the platform or the gaming elements that played the key (or more important) role in their work. A discussion on this aspect will strengthen their paper. Below are two recent papers that compare VR vs. a normal TV screen and may be helpful in this regard.

Xu et al. 2021. Effect of Gameplay Uncertainty, Display Type, and Age on Virtual Reality Exergames. In Proc. of the CHI'21, Article 439, 1–14. https://doi.org/10.1145/3411764.3445801.

Another aspect that is notable in their work is that all participants completed the 12-week experiment. This is not often so common. For example, in this recent work (https://games.jmir.org/2021/4/e29330/) reporting an experiment with 31 participants (all university students), 15 completed a 6-week experiment. There are not a lot of details about their participants (e.g., graduate vs. undergraduate students, whether they regularly exercise, etc.). More details will help researchers to replicate their work with similar settings.

Reviewer 3 Report

Title: The Efficacy of an Immersive Virtual Reality Exergame Incorporating an Adaptive Cable Resistance System on Fitness and Cardiometabolic Measures: A 12-Week Randomized Controlled Trial

This study aimed to assess changes in physical performance, body composition, cardiometabolic health, participant training efficiency, enjoyment, and perceived exertion during a 3-month workout protocol and to compare differences between an Immersive Virtual Reality exergaming system and a self-directed conventional resistance training control.

Main comments

In general, the figures are not aligned with the text and the manuscript is well-written, however some specific comments are presented below.

1. Introduction

Line 41: “… allowing participants TO INTERACT AND to be ‘immersed’ in virtual environments in real-time CREATING A FEEDBACK”

2. Materials and Methods

Line 71: Delete “by word-of-mouth”: Colloquial expression.

After 2.1 point or 2.2 point the presence of a “study flow chart/diagram” figure could be visual to an easily understanding of this part.

3.Results

Line 282: Delete “(14 females)”: Data present in Table 1.

Table 1: Review and correct appropiated changes:

- Body fat (%) BBIVR: -3.8

- Fat mass (kg) SELF: -1.2

- SP 1-RM (kg) BBIVR: 28.6

- SP 85% 1-RM (reps) SELF: 2.0

- rVO2 max SELF: 0.88

- * Sit-in-reach (cm) SELF: 3.0

- Diastolic-BP (mmHg) BBIVR: 0.0

- RMR (Kcal) BBIVR: 132

Figure 3 has poor quality, please add another of better quality.

4.Discussion

No comments.

5. Conclusions

No comments.

6. References

Lines 464 and 492: “Appl. Physiol. Nutr. Metab. Physiol. Appl. Nutr. Metab.” It is the correct journal abbreviation?

Line 482: Include the surnames of the authors in reference 10.

Line 497: There is no journal in reference 17.

Review the punctuation rules (for instance, line 529 “PloS One. 2013” or line 554 “Sport. 2020”).

Round 2

Reviewer 1 Report

I have re-reviewed the manuscript entitled "The Efficacy of an Immersive Virtual Reality Exergame Incorporating an Adaptive Cable Resistance System on Fitness and Cardiometabolic Measures: A 12-Week Randomized Controlled Trial" for consideration to be published in International Journal of Environmental Research and Public Health . It is a very interesting study that probably will encourage another authors to perform further studies. Congratulations to all the authors.